# Century-old chromatin architecture revealed in formalin-fixed vertebrates

Erin E. Hahn [1], Jiri Stiller[2], Marina R. Alexander[1], Alicia Grealy[1], Jennifer M. Taylor[2], Nicola Jackson [3], Celine H. Frere[3] & Clare E. Holleley [1] ✉

Gene expression is regulated by changes in chromatin architecture intrinsic to cellular differentiation and as an active response to environmental stimuli. Chromatin dynamics are a major driver of phenotypic diversity, regulation of development, and manifestation of disease. Remarkably, we know little about the evolutionary dynamics of chromatin reorganisation through time, data essential to characterise the impact of environmental stress during the ongoing biodiversity extinction crisis (20th–21st century). Linking the disparate fields of chromatin biology and museum science through their common use of the preservative formaldehyde (a constituent of formalin), we have generated historical chromatin profiles in museum specimens up to 117 years old. Historical chromatin profiles are reproducible, tissue-specific, sex-specific, and environmental condition-dependent in vertebrate specimens. Additionally, we show that over-fixation modulates differential chromatin accessibility to enable semi-quantitative estimates of relative gene expression in vertebrates and a yeast model. Our approach transforms formalin-fixed biological collections into an accurate, comprehensive, and global record of environmental impact on gene expression and phenotype.

Chromatin, the cell's intricate web of DNA wrapped around proteins, orchestrates gene expression, shaping cellular identity and dynamically responding to ever-changing signals. Chromatin compaction, or architecture, exists on a spectrum from tightly packed (typically transcriptionally silent) heterochromatic regions to the more open and accessible (typically transcriptionally active) euchromatic regions. Chromatin architecture therefore provides clues as to which genes are being expressed at a given time or coincident with a given set of environmental conditions[1,2]. Characterising chromatin architecture changes throughout an organism's life can reveal functional regulatory mechanisms involved in development[3], aging[4], disease[5] and plastic response to environmental stressors such as changing climates. Reconstruction of palaeolithic epigenomes has provided temporal insight into gene regulation on deep evolutionary time scales[6–8]. However, changes in chromatin architecture in response to relatively rapid environmental changes occurring over the course of the last two centuries is limited by a mismatch between current molecular capability (e.g., ref. 9) and the state of preservation of historical specimens[10].

The field of chromatin biology was born in the mid-20th century, with methods relying upon the chemical fixative formaldehyde to preferentially cross-link histone-associated DNA and then characterise the chromatin landscape. Formaldehyde is still essential in modern techniques such as Micrococcal Nuclease (MNase) treatment[11,12], Formaldehyde Assisted Isolation of Regulatory Elements (FAIRE)[13], Chromatin Immunoprecipitation (ChIP)[14], and High-throughput Chromosome Conformation Capture (Hi-C)[15]. In parallel, from the early 1900s, the use of a formaldehyde-based media called formalin (3.7% formaldehyde), came into common use in histopathology, anatomy, and embalming human remains. Formalin media was also used extensively by early naturalists to preserve voucher specimens, facilitate detailed anatomical descriptions, and document biodiversity.

[1]National Research Collections Australia, Commonwealth Scientific Industrial Research Organisation, Canberra ACT 2601, Australia. [2]Agriculture and Food, Commonwealth Scientific Industrial Research Organisation, St Lucia, Queensland 4067, Australia. [3]School of the Environment, University of Queensland, St Lucia, Queensland 4067, Australia. ✉e-mail: clare.holleley@csiro.au

Thus, many of the earliest collected vertebrate specimens (including taxonomic "type" specimens) have been exposed to formaldehyde. Formaldehyde preservation is most common amongst taxa that do not have alternative means of preservation (e.g., fish, amphibians, reptiles) but is also applied regularly to all biota.

Although formaldehyde is commonly used to gather insight on chromatin conformation and DNA is regularly recovered from formalin-fixed, paraffin-embedded (FFPE) tissue blocks[16,17], a common dogma has persisted among molecular biologists and museum curators that formaldehyde prohibits molecular work in museum specimens. Recently, this preconception has begun to shift, thanks to studies demonstrating the successful recovery of historical DNA from formaldehyde-fixed museum specimens[18–23]. Still, translating chromatin assays for use in museum specimens requires overcoming additional challenges.

Museum preservation practices present several obstacles for chromatin biology applications: (1) variation in post-mortem decomposition prior to fixation results in variable biomolecular degradation among specimens; (2) fixation of whole specimens may result in uneven fixation across tissues, and; (3) exposure of museum specimens to higher concentrations of formaldehyde (3.7% versus 1%) for longer periods of time (days versus minutes) with some specimens remaining exposed to formaldehyde indefinitely, results in heavier fixation compared with histological specimens. Additional challenges exist because fixation conditions for museum specimens are rarely (if ever) recorded in specimen metadata (e.g., fixation time/volume per gram), resulting in specimens having unknown preservation states. For this reason, previous studies reporting successful DNA extraction from formalin-preserved museum specimens have either relied on incomplete metadata or simply assumed that wet collection items of a certain age used formalin. By assessing specimen quality directly through visual inspection of the tissue and measurement of media pH and formaldehyde concentration, we have previously shown that formalin-preserved specimens can be identified and vetted for sequencing suitability[18,24]. Thus, a robust and customised approach is required to characterise historical chromatin architecture in the unique context of extreme fixation and DNA degradation in museum specimens.

In this work, we utilise the common application of formaldehyde fixation in both modern chromatin biology and museum science to generate historical chromatin profiles, thus, unlocking a century's worth of gene regulation data. We adapt two formaldehyde-based chromatin accessibility assays, FAIRE-Seq and MNase-Seq. Specifically, FAIRE-Seq enriches for open chromatin (i.e., euchromatin) by using phenol/chloroform extraction to remove formaldehyde-crosslinked nucleosome-associated DNA (i.e., heterochromatin), whereas MNase-Seq enriches for nucleosome-bound DNA through enzymatic digestion of euchromatin. We test the hypothesis that chromatin architecture is preserved in formaldehyde-exposed historical specimens, observable as sequence-read depth variation associated with chromatin accessibility. Preliminary reports hinted at this potential, due to read-periodicity patterns observed in shotgun whole genome sequencing data from formaldehyde-preserved museum specimens[18] that resembled the signature of nucleosome positioning observed in ancient (>4000 years) DNA[7]. When optimised, archival FAIRE and MNase assays could offer species-agnostic antibody-free characterisation of chromatin accessibility across eukaryotes.

## Results

### Understanding heavy fixation in a yeast model
We conducted initial optimisation using a fixation time series in a well-characterised experimental yeast system (*Saccharomyces cerevisiae*). This series established the molecular consequences of over-fixation on visualising chromatin accessibility. We cultured yeast under optimal and heat shock conditions, measured expression differences from fresh cells via RNA-Seq and sequenced DNA recovered from cells fixed

with 1% formaldehyde for 15 min, 1, 6 and 24 h processed with established FAIRE[25] and MNase[26] workflows. Then, we called accessibility signals as occupancy values in DANPOS3[27] and tested for significant peak width changes (FDR < 0.05) between assay and input control. We observed an assay-specific progressive shift in the abundance and morphology of occupancy signal using both FAIRE and MNase methods (Fig. 1A, Supplementary Fig. 1A). Fixation-induced changes in occupancy morphology were most evident in the MNase assay, which also had a higher reproducibility across three technical replicates (genes shared between replicates: MNase = 77–83%; FAIRE = 7–21%; Fig. 1B, Supplementary Fig. 1B). This difference in assay sensitivity and reproducibility is consistent with modern studies that show that FAIRE consistently has a low signal-to-noise ratio compared to other assays[25,28]. Interestingly, fixation time had no significant impact on the proportion of genes shared between replicates in either assay, indicating that fixation alters but does not destroy occupancy signals (Supplementary Fig. 1C).

Genome-wide chromatin accessibility profiles were informative about the regulatory response of yeast to heat shock under all fixation conditions but were most definitive at the two extremes (15 min versus 24 h fixation). Using MNase, chromatin occupancy shifts successfully identified the directionality of expression changes in response to heat shock (established by RNA-seq) (Fig. 1C). Of the 383 genes with MNase signal gain after 24-h fixation, 108 were also identified as upregulated by RNA-Seq, which is significantly higher than the mean number of genes shared between two random gene sets across 1000 permutations (Fig. 1E; mean = 19.989, standard deviation = 4.37, adjusted $p$-value = 6.061912$^{-87}$). GO term enrichment analysis with EnrichR[29,30] identified terms associated with heat stress, with two of the top five GO terms identified in heavily fixed chromatin being shared with terms identified via RNA-Seq (Fig. 1F). Additionally, under maximal fixation conditions (24 h), the MNase assay displayed a significant positive correlation between the magnitude of occupancy shifts and the magnitude of transcriptional activity ($R = 0.52$, $p < 0.001$; Fig. 1D, Supplementary Fig. 2). Thus, over-fixation of chromatin may provide semi-quantitative information about gene expression, an advance over-and-above the existing utility of modern non-quantitative assays.

### Assay adaptation to museum specimens
Having established that the chromatin accessibility state is recoverable despite excessively long fixation conditions in yeast, we then adapted both assays for use in heavily fixed archival vertebrate tissues based on established protocols for fresh vertebrate tissues[25,31]. Significant optimisation was required to dissociate fixed multicellular tissue, improve the recovery of highly degraded and heavily crosslinked chromatin, and thus enhance the modulated archival chromatin architecture signal (Fig. 2). To robustly develop and test our archival assays under standardised conditions, we created an experimental collection of formalin-preserved inbred C57 Black 6 laboratory mice and outbred wild-trapped mice (*Mus musculus*) with specimen-matched flash-frozen liver tissue (Table 1).

By exploiting the properties of over-fixation, our MNase and FAIRE protocols successfully enriched for occupancy signal changes in regions ±2 kb of transcription start sites (TSSs, Fig. 3A, C, Supplementary Figs. 3A, B, 4 and 5) that produced occupancy profiles matching to the tissue of origin (Fig. 3E, Supplementary Fig. 3C). Notably, the MNase assay of archival mouse tissue signal showed stronger enrichment globally at TSSs compared to FAIRE (Supplementary Fig. 3A), a higher degree of repeatability between replicates (53% overlap in genes identified in all three laboratory mouse replicates compared to 31%; Supplementary Fig. 3B), and the MNase gene sets more closely resembled those from fresh tissue compared to the FAIRE assay (51% agreement compared to 38% in laboratory mice; Supplementary Fig. 3C). Here, we focus on the archival MNase assay results, however, further exploration of the relative sensitivities of the

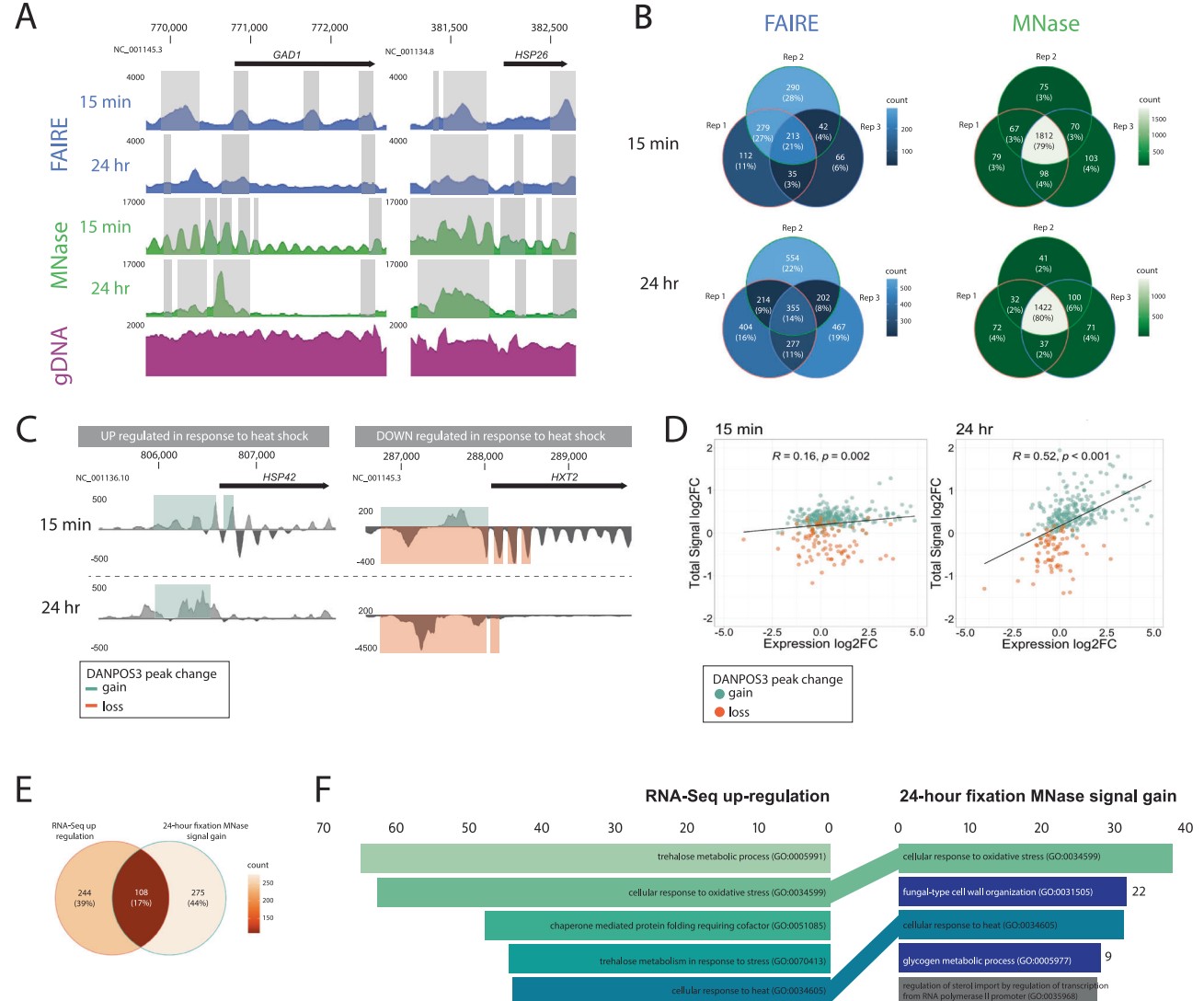

**Fig. 1 | Heavy formaldehyde fixation modulates but does not eliminate chromatin architecture evidence in experimental yeast cultures. A** Pooled occupancy values (FAIRE: blue, MNase: green) compared to gDNA extraction control (purple) with 15 min or 24 h fixation of heat-shocked *Saccharomyces cerevisiae*. Shading indicates regions with significant peak width shifts (FDR < 0.05). Upstream of highly upregulated *GAD1* and *HSP26* genes (log2FC = 3.8 and 9.02), changes in occupancy signal morphology are observed (y-axis) relative to the genomic position (x-axis). The 5′ FAIRE peak broadens, while the distinct 5′ MNase nucleosome array transforms into a single peak. **B** Venn diagrams demonstrate the repeatability of FAIRE (blue) and MNase (green) treatment among technical replicates. Numbers/proportions represent genes with significant peak gain (FDR < 0.05, log10Pval < −6) within 2 kb upstream of the transcription start site (TSS). **C** Differential DANPOS3 occupancy values were calculated by comparing pooled heat-shocked and optimal growth replicates treated with MNase. Statistical significance was assessed using a two-sided *t*-test, and the FDR was controlled using the Benjamini−Hochberg procedure to adjust for multiple comparisons. Signal changes are shown for a highly upregulated gene (*HSP42*, log2FC = 4.86) and a highly downregulated gene (*HXT2*, log2FC = −2.21) as measured by RNA-Seq in fresh cultures. Shading indicates

regions of significant (FDR < 0.05, log10Pval < −6) peak gains (green) or losses (orange). **D** Total signal log2FC for genes with significant (FDR < 0.05) total peak signal change between pooled replicate heat shock and optimal growth conditions in the 2 kb region upstream of the TSS is plotted against expression log2FC. Genes shared between the 15-min and 24-h time points are shown (green = signal gain; orange = signal loss). Linear regression lines are fitted with correlation coefficients (R) and *p*-values. **E** Venn diagram illustrating the overlap between genes identified as upregulated via RNA-Seq and with MNase peak gains in yeast fixed for 24 h. **F** Gene Ontology (GO) Biological Process enrichment: Genes with significant peak gain across pooled replicates (FDR < 0.05, log10Pval < −6) within 2 kb upstream of the TSS in MNase-treated yeast fixed for 24 h (N = 383) compared to significantly upregulated genes (N = 352) measured by RNA-Seq. GO term enrichment was calculated using Enrichr, with statistical significance assessed using a Fisher exact test. The FDR was controlled using the Benjamini−Hochberg procedure for multiple comparisons. The length of coloured bars corresponds to the Enrichr combined score [log(*p*-value) * *z*-score]. MNase GO terms are coloured shades of green, dark blue or grey if they are found in the top 5, top 25 or not within the RNA-Seq GO terms. Source data are provided as a Source Data file.

archival MNase and archival FAIRE assays may reveal additional insight into the effects of archival fixation on chromatin architecture as well as providing parallel lines of evidence to characterise historical gene regulation.

Critically, the genome-wide signature of chromatin architecture in aged multicellular museum specimens (as opposed to yeast cultures) is the inverse of the standard MNase and FAIRE profiles from minimally

fixed fresh tissue (Fig. 3A, Supplementary Fig. 3A). This indicates that the archival assays work to reveal historical chromatin accessibility through depletion of open active chromatin rather than through enrichment, a unique property of the archival assay. To explain the inverse occupancy signal, we propose a model under which fixation, cellular dissociation, and age of specimen influence chromatin accessibility and thus regional enrichment or depletion (Fig. 4). This model

**Fig. 2 | Overview of the archival chromatin assay workflow.** We prepare heavily fixed archival tissue nuclei for chromatin extraction through a stepwise process. This includes cryo-pulverisation for tissue fracturing, enzymatic digestion with pepsin to improve dissociation, Dounce homogenisation for fine tissue disruption, and prolonged sonication (Nuclei Extraction by SONication). The tissue can then be processed via: Formaldehyde Assisted Isolation of Regulatory Elements (FAIRE) treatment with further sonication to shear the chromatin followed by reservation of a fraction for input control and phenol:chloroform extraction of the FAIRE fraction or Micrococcal Nuclease (MNase) treatment of the nuclei with co-digestion with MNase and Exonuclease III. Isolated chromatin then undergoes RNase and proteinase K treatment before DNA fragments are purified using phenol:chloroform extraction and SPRI bead purification optimised for small fragment recovery. Sequencing libraries are prepared using an IDT xGEN cfDNA & FFPE DNA kit for paired-end sequencing. Sequencing reads are mapped using kalign without prior trimming. Alignments are de-duplicated using unique molecular identifiers (UMIs) and undergo GC-bias correction. Enrichment analyses are performed using the DANPOS3 dpeak function. Prior to downstream analysis, confirmation of the expected peak loss versus peak gain pattern can be performed. Created with BioRender.com, released under a Creative Commons Attribution-NonCommercial-NoDerivs 4.0 International license.

unifies our mouse and yeast observations and demonstrates that the archival MNase assay is informative across single and multicellular eukaryotes.

Similar to the 24 h-fixed yeast data, the magnitude of occupancy change in our archival MNase assay appears to be a semi-quantitative proxy for gene expression (Fig. 3B, Supplementary Fig. 6). Highly expressed liver-specific genes (e.g., *APOC1*, Fig. 3B) had a strong

depletion signal in archival tissues whereas genes expressed at low levels in liver had no signal and did not vary from the input control (Supplementary Fig. 7). In both fresh and archival laboratory mouse liver tissue, we observed significant MNase signal changes at 60−75% of genes with high expression (zFPKM > 2) compared to approximately 25% of genes with low expression (zFPKM < −2) (Supplementary Fig. 6A). Unsurprisingly, given the FAIRE assay's low signal-to-noise

**Table 1 | Specimen details for mock-preserved mouse specimens**

| Sample name | ANWC Reg. no. | Genetic background | Time interval (years) | Sex | Weight (g) | Length (cm) |
|---|---|---|---|---|---|---|
| Lab1 | M37810 | C57BL6 | 4.8 | M | 23.5 | 9.4 |
| Lab2 | M37811 | C57BL6 | 4.8 | M | 22.7 | 9.5 |
| Lab3 | M37816 | C57BL6 | 4.8 | M | 23.6 | 9.7 |
| Wild1 | M37959 | Wild-Murrumbateman | 2.3 | M | 14.0 | 7.5 |
| Wild2 | M37971 | Wild-Murrumbateman | 2.3 | F | 19.5 | 8.5 |
| Wild3 | M37976 | Wild-Yass | 2.3 | F | 20.0 | 8.5 |

We processed six *Mus musculus* individuals for inclusion in a mock-preserved experimental specimen set. For each specimen, we provide the Australian National Wildlife Collection (ANWC) Registration number, the strain or collection site as the Genetic Background, the Time Interval in years between preparation and fixed-tissue sampling, and the individual's sex, weight, and length.

ratio, signal changes showed a relatively weaker and more variable association with gene expression 25–80% of genes with high expression (zFPKM > 2) and 0–20% of genes with low expression (zFPKM < −2) (Supplementary Fig. 6B). The consistency of relative archival assay performance (FAIRE versus MNase) with expectations derived from modern studies stands as further evidence that our assays can indeed be used to characterise historical chromatin architecture.

### Historical chromatin signatures

As the final demonstration of our approach, we characterised archival chromatin architecture in truly historical formalin-preserved museum specimens obtained from the Queensland Museum. We selected five eastern water dragon (*Intellegama lesueurii lesueurii*) specimens preserved in formalin between 1905 and 2001 (Table 2). Real museum specimens are a finite and precious resource, thus we only had liver tissue volume (29–200 mg) sufficient for a single archival chromatin assay per individual. We selected archival MNase due to its stronger occupancy signal, superior repeatability, and semi-quantitative association with gene expression. We note that whilst FAIRE was not conducted on these samples, it is still a valuable tool for future independent verification of MNase results. Given that we expected to achieve relatively low coverage from whole genome sequencing of the archival tissues, we also sequenced a modern fresh tissue genomic DNA extraction from the liver as an input to control for sequencing platform-specific technical biases[32].

All five water dragon samples produced clear evidence of genome-wide occupancy signal changes in regions ±2 kb of TSSs (Supplementary Fig. 8). These profiles validate our historical chromatin accessibility assay in specimens up to 117 years old. Due to their age, we lacked specimen-matched fresh tissue for the water dragons, which limits our ability to measure archival and fresh signal correlation in this system. In the absence of fresh tissue for comparison, we use our experimental mouse data for context to clearly observe that functional inference of historical gene regulation is possible, even with the small sample sizes in this study. For both vertebrate systems, the predominant occupancy signal is associated with phenotypic sex (Fig. 5A, C). Males and females cluster along the first PC axis using a Pearson correlation analysis, which explains a large proportion of the variation in chromatin profiles (Fig. 5A; Mouse PC1 = 97.8%; Water dragons = 76.5%). Interestingly, the magnitude of correlation within the water dragon analyses is roughly 20-fold that of those within the mouse analyses (Fig. 5C). This result may reflect expected differences in homoeothermic mammals with genetic sex determination compared to ectothermic reptiles exhibiting epigenetically regulated environmental sex determination[33]. Alternatively, it could be consistent with the occurrence of somatic cell-autonomous sex identity, a phenomenon previously observed in some birds and reptiles[34–37] but not in mammals.

Eliminating the strong influence of sex in PC1 and instead comparing PC2 and PC3, both mouse and water dragon chromatin profiles segregate into groups consistent with habitat type at the time of collection (Fig. 5B). Specifically, water dragons collected in urban

Brisbane cluster separately to individuals from non-urban bushland habitats; and lab mice form a tight, highly reproducible cluster quite distinct from the more genetically and transcriptionally diverse outbred wild mice. Increased sampling will allow investigation of the degree to which this signal is influenced by genetic similarity or population structuring. Our findings underscore the importance of sex-matching when selecting individuals for future work aiming to measure environmental effects. Future studies may also consider strategic selection of somewhat older formalin-preserved specimens having corresponding frozen tissue to confirm the correspondence of the archival chromatin signal with that from fresh tissue in specimens older than our relatively recently preserved mice.

Lastly, our water dragon results recapitulate previous results from our group that indicate that specimen age is a poor predictor of sequencing suitability[18]. Here, the 1905 specimen represents the oldest confirmed formalin-preserved museum specimen to successfully yield genome-wide sequencing data to date and the MNase reads alone yielded the highest whole genome coverage yet achieved in archival formalin-preserved specimens (genome cover = 5-8X; Table 3). Thus, our archival MNase method is suitable for obtaining both genomic and epigenomic data from formalin-preserved specimens and does so simultaneously.

## Discussion

Our perspective on the utility of formaldehyde-fixed archival specimens provides historical epigenetic capability targeting organisms collected during the 20th–21st century. Our methods open the door to systematic and comprehensive investigations into the temporal dynamics of chromatin accessibility by drawing upon the untapped potential of museums and global biorepositories. Contrary to the prevailing dogma, we have shown that over-fixation with formaldehyde does not destroy DNA but rather enables the successful recovery of historical chromatin architecture. Localised chromatin accessibility can be highly correlated with gene expression[38]. Thus, functional inference with our approach may be more powerful than differential DNA methylation analyses because of the inconsistent relationship of methylation with gene activity[39]. Thus, this capability has the potential to revolutionise the power of modern epigenome-wide association studies in the pursuit of functional regulatory variants by characterising vertebrate chromatin architecture over the last century.

Broad adoption of archival chromatin profiling techniques by the world's natural history collections and their users will require careful sampling designs to control for the effects of post-mortem degradation, specimen sex, genetic background, and age. While the interval between death and fixation is rarely if ever recorded, our group has previously reported that the integrity of the gut contents can be used as a proxy for degradation when vetting specimens[18]. Sex-specific gene expression is observed across a wide range of vertebrate tissues, even those that are not gonadal in origin or associated with secondary sexual characters[40]. Thus, controlling for sex should be a key consideration in any study design. Likewise, age of the individuals should be considered, given expected changes in chromatin accessibility

associated with aging[41,42]. We observed greater MNase signal variation in the wild mouse samples compared to laboratory mice, likely due to a combination of factors, such as variation in sex, age, diet, exercise, or genetic background. This indicates that a higher degree of replication

will be required within carefully matched specimen sets to study historical wild populations.

Now that recovery of genomic data from formaldehyde-fixed museum specimens has been firmly established by this and other

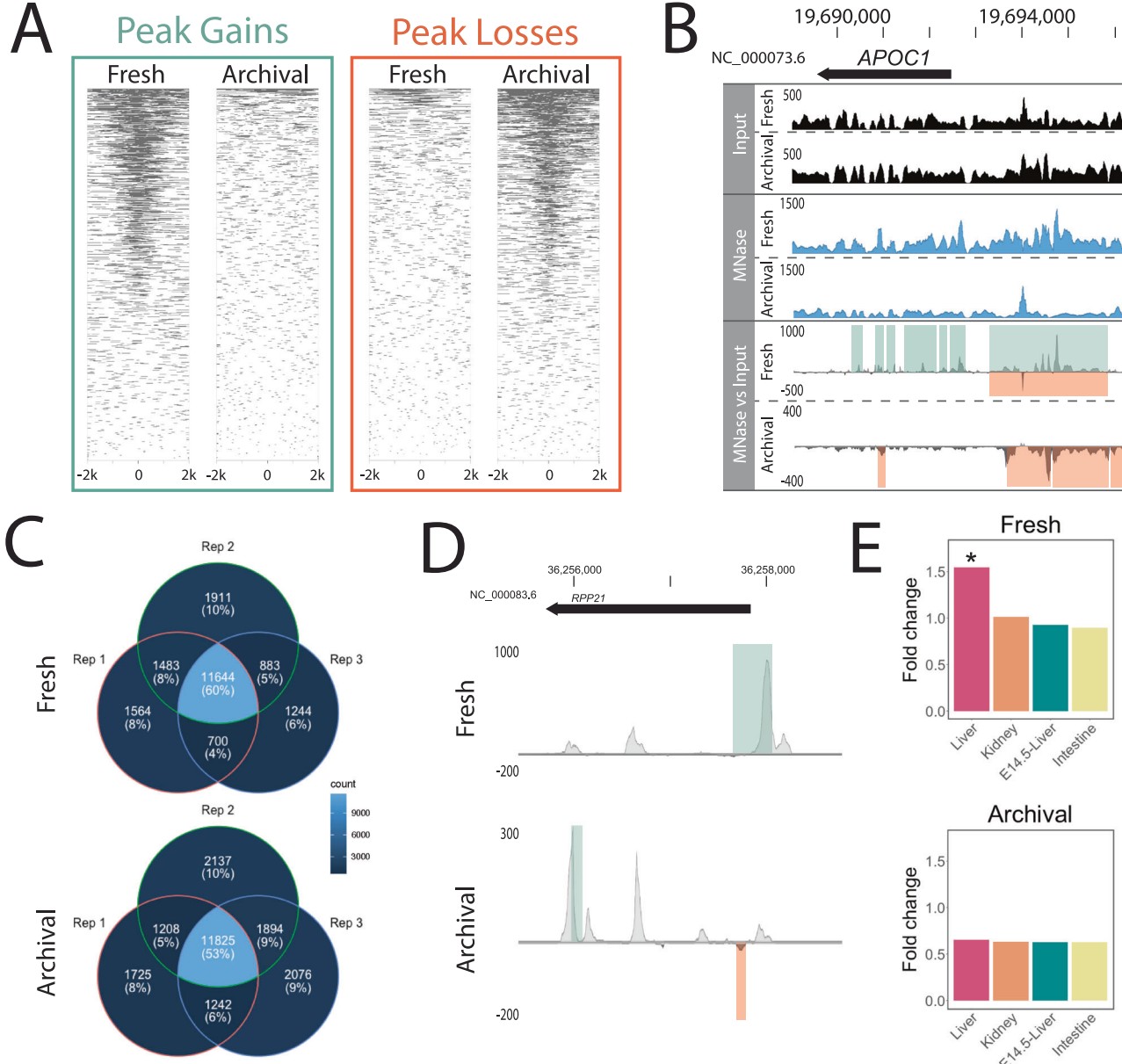

**Fig. 3 | Genome-wide occupancy profiles using MNase in archival mouse specimens are the inverse of freshly collected specimens. A** Heatmap of MNase assay significant peak gains and losses (FDR < 0.05, log10Pval < −6) in fresh and archival tissues 2 kb on either side of genome-wide transcription start sites (TSS) pooled across three individuals. **B** Pooled occupancy values as wiggle traces (DANPOS3 dpeak function) for input (black) and MNase (blue) as well as differential MNase signal over input control (grey) for fresh and archival *Mus musculus* liver tissue. Occupancy values and signal changes are shown upstream of a gene highly expressed in liver (*APOC1*, FPKM = 38,660) as measured by RNA-Seq in fresh tissue. Green and orange shading upon the differential signal panel represents significant (FDR < 0.05, log10Pval < −6) peak gains or losses across three individuals detected by DANPOS3. **C** Venn diagrams demonstrate the relative repeatability of the MNase assay applied to fresh and archival liver tissue among biological replicates in laboratory mice. Numbers/proportions represent genes with significant peak gains for fresh tissue and losses for archival tissue (FDR < 0.05, log10Pval < −6) within 2 kb upstream of the TSS. Lighter colours indicate higher shared gene count. **D** Differential pooled DANPOS3 occupancy values

comparing laboratory strain to wild-caught mice in fresh and archival liver tissue treated with MNase. Signal change is shown for a gene highly upregulated in laboratory versus wild mice (*RPP21*, log2FC = 9.611) as measured by RNA-Seq analysis of fresh tissue. Green and orange bars represent significant (FDR < 0.05, log10Pval < −6) peak gains or losses detected by DANPOS3. **A–D** Statistical significance of peak loss or gain was assessed using a two-sided *t*-test, and the FDR was controlled using the Benjamini–Hochberg procedure to adjust for multiple comparisons. **E** Genes with pooled occupancy signal changes (Fresh = gains; Archival = losses) show highest enrichment (fold change over a set of all mouse protein-coding genes) for genes expressed in the liver in both fresh and archival mouse tissues. For each panel, shared gene lists were assembled from a pool of three laboratory and three wild mice and enrichment within Mouse ENCODE datasets was calculated with TissueEnrich. Statistical significance of enrichment was assessed using a hypergeometric test, and the FDR was controlled using the Benjamini–Hochberg procedure.* Significant (Benjamini–Hochberg adjusted *p*-value < 0.001) enrichment above background. Source Data are provided as a Source Data file.

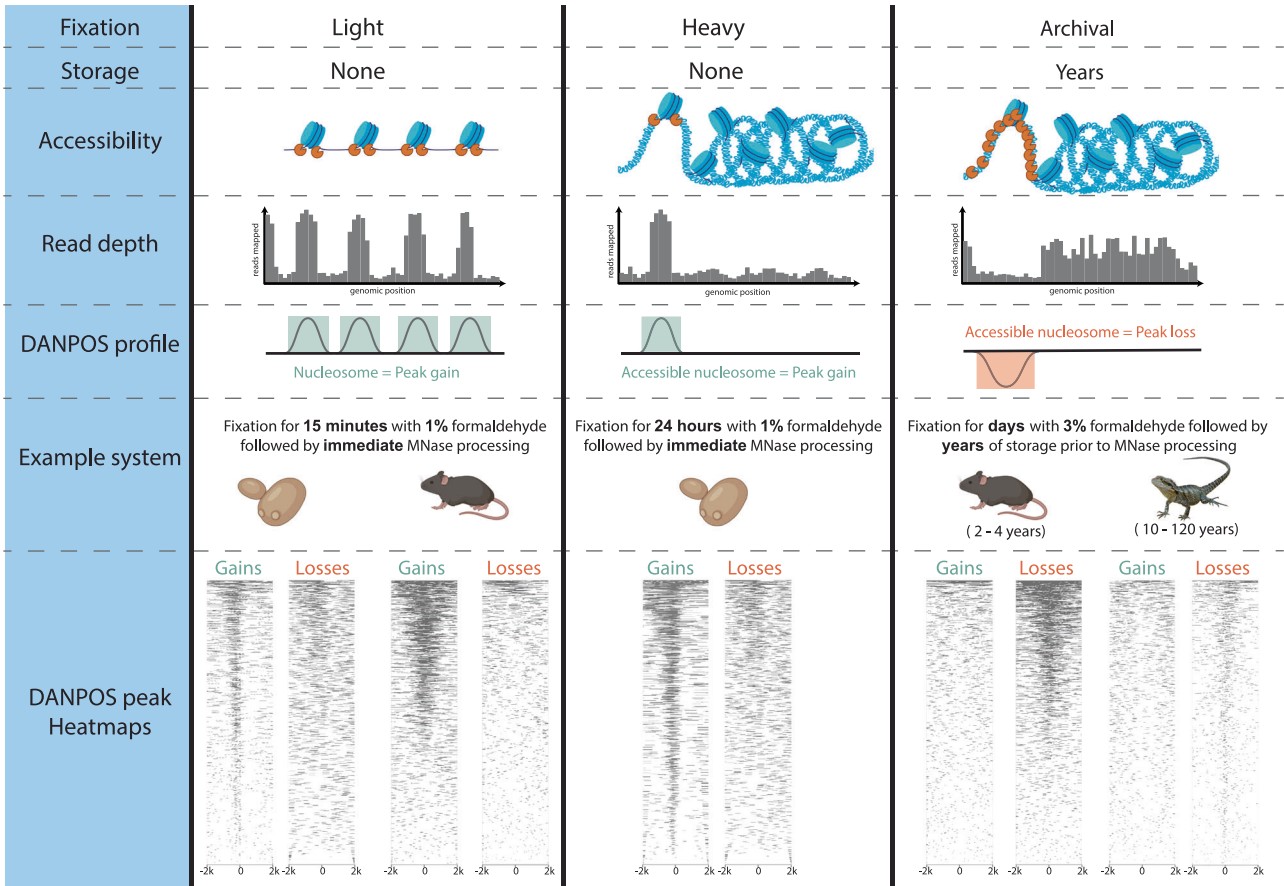

**Fig. 4 | Proposed model of the effect of fixation and long-term storage on MNase accessibility and occupancy signal.** Conceptual model of the combined effects of fixation and storage conditions on MNase occupancy signal. From top to bottom, **(Accessibility)** Applied to lightly fixed chromatin, the MNase enzyme (depicted in orange) cleaves DNA adjacent to the nucleosome and resects unbound DNA, thus releasing nucleosome-bound DNA. Applied to heavily fixed chromatin, the MNase enzyme's access to unbound DNA is modulated by chromatin accessibility, thus reducing the release of nucleosome-bound DNA only from within the most accessible chromatin regions. Applied to archivally fixed chromatin, prolonged MNase digestion is required to release sufficient DNA for sequencing from the heavily fixed chromatin within intact whole specimens stored for months to many years. This prolonged digestion preferentially degrades both linker DNA and nucleosome-bound DNA in MNase-accessible regions and releases fragments from relatively inaccessible regions. **(Read depth)** Relative accessibility of the MNase enzyme alters the read depth pattern observed in the region of euchromatin relative to heterochromatin. **(DANPOS profile)** DANPOS efficiently detects both relative occupancy value gains and losses resulting from MNase digestion. **(Example system)** We offer examples of light fixation in both a single (yeast) and multicellular (mouse) system with no storage time, heavy fixation in a single (yeast) cellular system with no storage time and archival fixation in two multicellular vertebrate systems stored for several years (mouse) or up to 117 years (water dragon). **(DANPOS peak Heatmaps)** For each example system, we show a heatmap of MNase assay significant peak gains and losses (FDR < 0.05, log10Pval < −6) 2 kb on either side of genome-wide transcription start sites pooled across all replicates (three for yeast and mouse, 5 for water dragon). Under light and heavy fixation, the predominant genome-wide signal appears as occupancy gains, while under archival fixation, the predominant genome-wide signal appears as occupancy losses irrespective of storage time. Created with BioRender.com, released under a Creative Commons Attribution-NonCommercial-NoDerivs 4.0 International license.

studies[18–23], we should reassess our assumptions about the damaging effect of fixation on other nucleic acids and epigenetic modifications. For example, could quantitatively informative mRNA feasibly be retrieved from historical specimens? Precedents have been set by successful research on clinical FFPE samples[43], and a single study has recently retrieved RNA from formalin-fixed museum specimens[44]. A better understanding of how fixation modulates the molecular signal could revolutionise our ability to study long-term temporal trends in gene regulation. Until now, elucidating the effects of fixation on molecular signals in an archival context has been limited by a lack of specimen-matched fresh-frozen tissues with which to validate archival signatures. Furthermore, and perhaps more importantly, preservation procedures and reporting lack standardisation, leading to variation and uncertainty in the fixation status of specimens.

To develop methods tailored to the challenges posed by archival specimens, we created a bank of mock-preserved specimens with matched frozen tissue and extensive preservation metadata. We encourage the world's museum collections to follow suit and prepare methodologically focused collections for the purpose of enabling future research. Further to that, we recommend the inclusion of preservation history details among specimen metadata to enable accurate specimen suitability screening. Doing so will ensure that the methods described here become part of a suite of approaches to characterise historical genomes and gene regulation. Achieving this feat will increase the power of modern studies seeking to establish causal functional relationships between regulatory variation and phenotypes, providing a parallel to the transformational insight gained from ancient DNA genomic analyses which have identified functional human sequence variation[45].

Our method is likely to be appropriate for use across vertebrates, with the only requirement being the availability of a species-specific or phylogenetically related reference genome. Museum scientists and wildlife researchers may not yet routinely use chromatin accessibility data but should be aware of the capability this approach adds. For

**Table 2 | Specimen details for archival eastern water dragon specimens**

| Sample name | QM Reg. no. | Habitat | Latitude | Longitude | Time interval (years) | Sex | pH | [F] |
|---|---|---|---|---|---|---|---|---|
| 1905M | J91140 | Urban | −27.46666667 | 153.0166667 | 116 | M | 6.78 | 400 |
| 1990M | J51722 | Urban | −27.35 | 152.9666667 | 31 | M | 6.53 | 200 |
| 1992M | J54438 | Non-urban | −27.91666667 | 152.3333333 | 29 | M | 6.53 | 200 |
| 2001F | J76769 | Non-urban | −27.83333333 | 153.1666667 | 20 | F | 6.53 | 200 |
| 2001M | J76089 | Urban | −27.51666667 | 152.95 | 20 | M | 6.53 | 200 |

We selected five archival eastern water dragon (*Intellegama lesueurii lesueurii*) specimens for gDNA and MNase processing. For each specimen, we provide the Queensland Museum (QM) registration number, Habitat type, Latitude and Longitude of the collection location, the Time Interval in years between preparation and fixed-tissue sampling, Sex of the individual as well as pH and residual formaldehyde concentration ([F] as mg/L) of the specimen media at the time of sampling. Note: all collection localities are in Queensland, Australia.

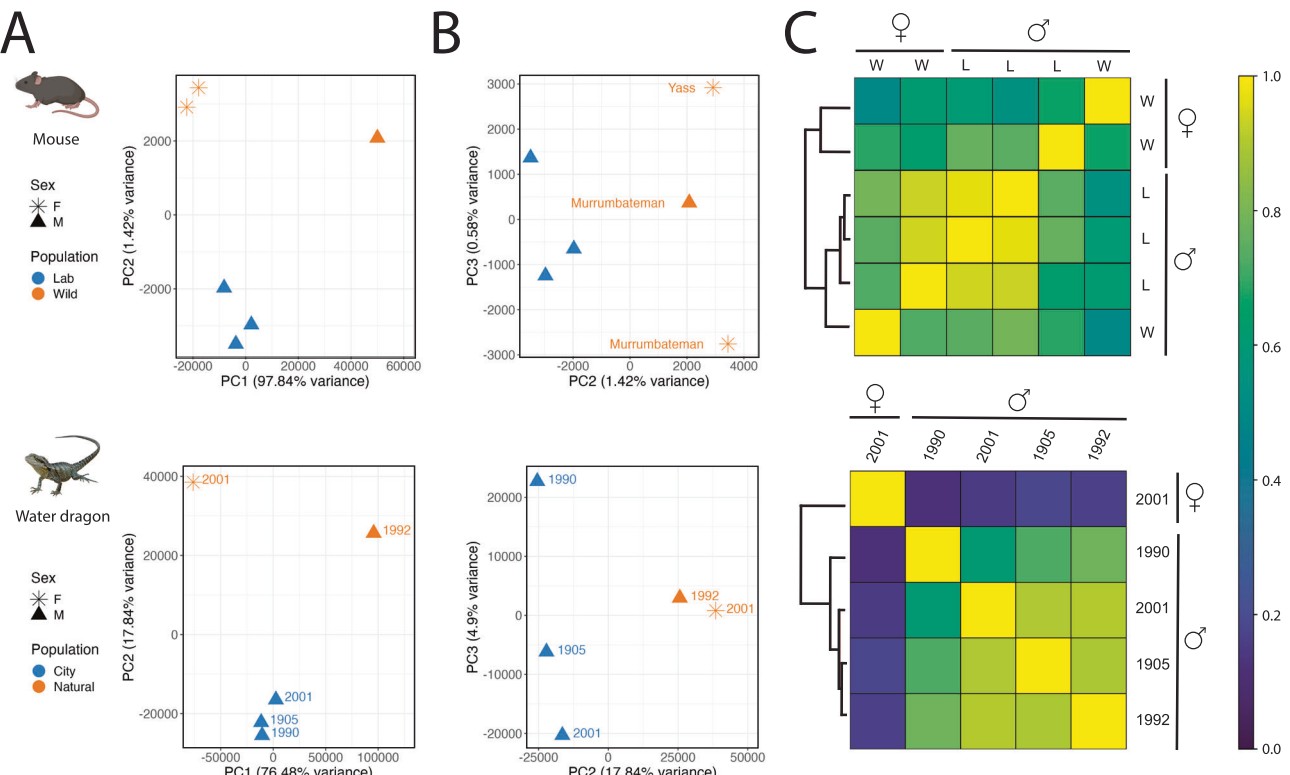

**Fig. 5 | Sex and population signatures in heavily fixed archival specimens.** In two species (mouse and water dragon), Pearson correlation-based analysis of genome-wide archival MNase occupancy signals resolves clusters of individuals by sex and population. PCA plots of principal components one and two (**A**) illustrate strong separation of females from males, while plotting components two and three (**B**) reveals clustering of individuals along PC2 in accordance with population. For mouse, only the autosomal chromosomes were considered for this analysis. The shape of individual PCA plot points indicates the specimen's sex (F = star; M = triangle) and colour indicates the population (mouse−blue = laboratory, orange =

wild; water dragon – blue = urban, orange = non-urban). Wild mice are labelled by collection location and water dragons are labelled by collection date.
**C** Representing the Pearson correlation scores of the first two PCA axes as a heatmap, a relatively stronger differentiation of the sole female water dragon from the four males emerges in comparison to the sex-based differentiation observed in mouse. In the mouse heatmap, as in the PCA plots, the individuals cluster first by sex and then by population (L = lab; W = wild). Created with BioRender.com, released under a Creative Commons Attribution-NonCommercial-NoDerivs 4.0 International license.

example, systematists can gain both the standard genomic data required for phylogenetic and/or population genomic analysis (e.g., whole mitochondrial genomes, orthologous gene sets, single nucleotide polymorphisms) and simultaneously characterise adaptive responses to the environment. This extra layer of information could help detect species provenance, identify structure in widely distributed vagile populations (e.g., fish stocks) or disambiguate rapid or recently diverged lineages (e.g., introduced and invasive species). Conservation biologists and wildlife managers may use the method to establish historical baselines against which to track rapid adaptation to captivity during ex-situ conservation efforts. Having a quantitative and reproducible molecular approach to characterise phenotypic plasticity allows managers to establish clear decision-making thresholds for

species release and red-flag triggers during ex-situ monitoring. There is also the potential to contribute to biosecurity monitoring either by the direct detection of pathogens and/or host responses to infection.

Most importantly, our method provides the unique opportunity to survey gene expression in both modern and historical specimens and reveal the mode, tempo and directionality of gene expression change over the last century. Generation of historical chromatin profiles at scale may enable statistical modelling approaches that leverage past and present data, to predict future state[10]. This provides essential context for estimating the resilience of contemporary populations to future environmental threats and may reveal the full repertoire of rapid evolutionary responses to environmental challenges such as climate change and introduced pests and pathogens.

**Table 3 | Extraction and sequencing details for the eastern water dragon specimens**

| Specimen | Archival tissue weight (mg) | DNA yield (ng/mg tissue) | Raw read pairs (M) | Reads mapping (%) | Mean insert size (bp) | Mean genome coverage (X) |
|---|---|---|---|---|---|---|
| 1905M | 82 | 0.08 | 146 | 40 | 74 | 5 |
| 1990M | 29 | 0.24 | 151 | 28 | 65 | 3.2 |
| 1992M | 45 | 0.12 | 171 | 32 | 68 | 4.3 |
| 2001F | 200 | 0.25 | 162 | 49 | 76 | 7.1 |
| 2001M | 92 | 0.63 | 154 | 58 | 78 | 8.1 |

Summary of the extraction and sequencing results for five archival eastern water dragon specimens processed with archival MNase treatment. For each specimen, we processed the available mass of archival liver tissue (reported in mg) and report DNA yield in ng/mg. For all tissues, we report DNA yield in ng/mg of tissue, the number of raw read pairs (millions), the percentage of raw reads mapping to the reference genome, the mean insert size of mapped reads in base pairs and the mean genome coverage after de-duplication and GC correction.

## Methods

All animal research conducted in this study complies with the Australian Code for the Care and Use of Animals for Scientific Purposes (2013) and protocols were approved by the CSIRO Wildlife and Large Animal Ethics Committee and the University of Sunshine Coast Animal Ethics Committee. The Australian National Wildlife Collection and the Queensland Museum provided permission to sample mouse and eastern water dragon museum specimens. All reagent sourcing is listed in Supplementary Data 1.

### Yeast processing

We used established procedures for MNase[26] and FAIRE[25] with yeast and made optimisations for processing heavily fixed cultures.

**Culture, fixation, and preparation of nuclei.** We grew *Saccharomyces cerevisiae* strain BJ5464 auxotroph ΔURA3 in 500 mL Yeast Extract-Peptone-Dextrose (YPD; 1% w/v yeast extract, 2% w/v peptone, 2% w/v d-glucose) medium to OD600 = 0.75 at 28 °C. We split the culture into equal volumes; one underwent 37 °C heat stress for 20 min. We collected 2 mL aliquots from both conditions for gDNA and RNA extraction.

We added formaldehyde (1%) to both optimal and heat shock flasks and incubated with slow shaking at room temperature. At 15 min, 1 h, 6 h and 24 h we collected aliquots and quenched fixation with addition of glycine to 0.125 M followed by gentle shaking at room temperature for 15 min. We pelleted the fixed cells (4000×g, 5 min, 4 °C), washed twice with Phosphate Buffered Saline, pH 7.4 (PBS), then resuspended in 10 mL cold Glycine Tris EDTA (GTE; 100 mM glycine, 10 mM Tris-HCl, pH 8.0, 1 mM EDTA) buffer and incubated at 4 °C with rocking for 24 h.

We isolated nuclei by harvesting fixed cells (4000×g, 5 min, 4 °C), washed twice with cold milliQ water, and resuspended in 2 mL Spheroplast Buffer (SB; 1 M Sorbitol, 50 mM Tris, pH 7.5 with freshly added 10 mM β-mercaptoethanol). We obtained spheroplasts by adding Zymolase 20T (MP Biomedicals) to 0.25 mg/mL and incubating at 28 °C for 2 h. We harvested fixed spheroplasts (1500×g, 10 min, 4 °C), washed twice with cold SB and resuspended in 10 mL. We aliquoted spheroplasts such that each aliquot contained a volume of cells from 4 mL and 10 mL of original culture for the short (15 min and 1 h) and long (6 and 24 h) cultures, respectively. We collected more cells from longer fixation time points to account for lower expected DNA recovery due to heavy fixation. We pelleted spheroplasts (1500×g, 10 min, 4 °C), removed the supernatant and froze the tubes at −80 °C.

**gDNA and RNA extraction from unfixed yeast.** We extracted gDNA and RNA from unfixed cells frozen at −80 °C. For gDNA, we resuspended cells in 200 μL SB, added Zymolase 20T to 0.25 mg/mL and 1

μL RNase A and incubated at 37 °C for 30 min. We harvested spheroplasts (3000×g, 10 min, 4 °C), resuspended in 100 μL PBS plus 0.01 M EDTA, added 2 μL of proteinase K (20 mg/mL) and incubated at 56 °C for 45 min in a thermal mixer with agitation at full speed (1400 rpm). We purified gDNA using phenol:chloroform:isoamyl alcohol (25:24:1) extraction and concentrated the gDNA on beads via small fragment-optimised bead purification (described below), eluting in 20 μL 10 mM Tris, pH 8.0. For RNA, we resuspended three aliquots per culture in 450 μL RLT Buffer and followed the manufacturer's instructions for the Qiagen RNeasy plant mini kit, eluting in 30 μL nuclease-free water.

**Small fragment-optimised bead purification of DNA.** We concentrated purified DNA with a custom small fragment-optimised SPRI bead clean-up. We prepared the bead solution in 50 mL aliquots from 1 mL Sera-Mag (Cytiva) beads such that when added to the DNA extract in a ratio of 1.5:1 (bead solution:DNA) the final concentration of reagents equal 12% PEG-8000, 40% isopropanol, 0.6 M NaCl, 6 mM Tris-HCl (pH 8.0), 0.6 mM EDTA and 0.03% Tween-20. After adding the bead solution, we incubated tubes at room temperature for 15 min with rotation, pelleted on a magnet, and removed the supernatant. We washed the beads with fresh 70% ethanol, pelleted them on a magnet and washed once more with 70% ethanol. After ethanol removal and brief drying for 30 s, we resuspended beads in 20 μL 10 mM Tris-EDTA and incubated at 37 °C for 15 min to thoroughly elute the DNA. Finally, we pelleted the beads on a magnet and transferred the supernatant to a new tube.

**Yeast MNase treatment.** We resuspended triplicate aliquots of each fixation timepoint (heat shock and optimal growth) in 200 μL MNase Digestion Buffer (DB; 0.5 mM spermidine, 0.075% Nonidet P40, 50 mM NaCl, 10 mM Tris pH 8.0, 5 mM MgCl₂, 5 mM CaCl₂ plus freshly added cOmplete EDTA-free protease inhibitor cocktail (Merck)), added 0.5 U of micrococcal nuclease (Worthington Biochemical Corporation) and incubated the tubes for 7 min in a 37 °C water bath. We quenched the reaction with 50 μL Quenching Solution (QS; 4% Triton X-100, 1.2% SDS, 600 mM NaCl, 12 mM EDTA) and incubated the tubes on ice for 10 min. We collected digested chromatin by centrifuging (14,000×g, 10 min, 4 °C) and removing the supernatant to a new tube. We digested the supernatant with proteinase K at a final concentration of 0.1 mg/mL and incubated at 56 °C for 2 h. We purified DNA with two phenol:chloroform:isoamyl alcohol (25:24:1) extractions, RNase treatment (1 μL RNase A and incubation at room temperature for 30 min) between extractions one and two followed by via small fragment-optimised bead purification, resuspending in 20 μL 10 mM Tris, pH 8.0 and fully de-crosslinking at 65 °C overnight.

**Yeast FAIRE treatment.** We resuspended four aliquots from each fixation timepoint (heat shock and optimal growth) in 1 mL Chromatin Shearing Buffer (CSB; 10 mM Tris-HCl pH 8.0, 0.1% SDS, 1 mM EDTA) and transferred each to a 1 mL Covaris milliTUBE. We sonicated the tubes in a Covaris E220 focused-ultrasonicator on settings PIP 420, duty factor 30%, cycles per burst 200 for 7 min (short fixation time points) or 12 min (long fixation time points). After clarifying the lysate by centrifugation (5500×g, 15 min, 4 °C), we set aside one tube per sample type at this point as an input control. With the remaining tubes, we performed phenol:chloroform:isoamyl alcohol (25:24:1) extraction with back-extraction of the organic phase with addition of 150 μL 10 mM Tris, pH 8.0 followed by an additional extraction. To the aqueous and input control tubes, we added 10 μL RNase A and incubated at room temperature for 30 min, followed by addition of 2 μL proteinase K and incubation at 55 °C for 1 h. We de-crosslinked overnight with incubation at 65 °C and concentrated the DNA via small fragment-optimised bead purification, resuspending in 20 μL 10 mM Tris, pH 8.0.

## Mock archival specimen preparation

To enable comprehensive methods development for this and future studies, we created a bank of experimental specimens representative of the wide variety of liquid preservation practices used in collections over the last century. We selected a species with a well-annotated genome and tractable tissue volume for processing and storage, *Mus musculus* (strain C57BL/6). We acquired 151 male mice aged 17–18 weeks from Australian BioResources and sacrificed them upon arrival by cervical dislocation (Australian Ethics Committee number 2017-34). In accordance with modern archival procedure, we sampled liver tissue from specimens for storage at −80 °C as specimen-matched fresh tissue. We then processed the intact carcasses using several preservatives (unbuffered formalin, neutral buffered formalin, and formalin acetic acid) with fixation times ranging from 15 min to 3 days with or without washing with water and stepping into ethanol for long-term storage. For all mice used in this study, we immersed each carcass in 10% neutral buffered formalin (3.7% formaldehyde) for 3 days followed by soaking in water for 1 day before transfer to 70% ethanol.

Additionally, the CSIRO Health and Biosecurity Rodent Management Team donated adult *M. musculus* live-trapped in May 2019 from two New South Wales locations (Murrumbateman, lat. −35.0424064, long. 148.99947; Yass, lat. −34.8682227, long. 149.00763) (Australian Ethics Committee number 2018-46). The wild mice had been housed for 3 months according to standard mouse husbandry practices at ambient temperature, humidity and natural day/light cycle prior to sacrifice by cervical dislocation. We processed these wild mice as we did the laboratory mice with sampling of fresh liver for cryogenic storage and immersion of each carcass in 10% neutral buffered formalin (3.7% formaldehyde) for 3 days, followed by soaking in water for 1 day before transfer to 70% ethanol.

We archived all mouse specimens in the Australian National Wildlife Collection (ANWC; Crace, ACT, Australia) spirit vault in glass specimen jars (see Table 1 for registration details of specimens used in this study).

## Vertebrate specimen selection and archival tissue sampling

**Archival mice.** We dissected archival liver tissue from mock specimens and transferred the tissue to ethanol-filled tubes for transport and further processing. At the time of dissection, the formalin-fixed laboratory and wild-caught mice had been archived for 4.8 and 2.3 years, respectively.

**Archival eastern water dragons.** Before sampling archival eastern water dragons (*Intellegama lesueurii lesueurii*), we assessed the sequencing suitability of specimens archived at the Queensland Museum. Following established methods[18], we took aliquots of the preservation media and measured pH and residual formaldehyde concentration. From visually well-preserved specimens within jars registering neutral pH (6 < pH < 8) and low formaldehyde ([F] < 10,000 mg/L), we sampled archival liver tissue and transferred the tissue to ethanol-filled tubes for transport and further processing.

## Archival vertebrate tissue processing

The following describes the final optimised tissue processing procedure we used on all archival specimens.

**Tissue preparation.** Following established methods[18], we cryopulverised the tissues into a rough powder using a cryoPREP automated dry pulveriser (Covaris) with three impacts to an extra thick TT1 TissueTube on intensity setting 6. We rehydrated the pulverised tissue under ice-cold conditions, stepwise by from 50% ethanol to water rocking at 4 °C for 10 min intervals. We collected tissue by centrifugation (4000×*g*, 5 min, 4 °C) and quenched residual formaldehyde by rocking overnight at 4 °C in an excess volume (approximately 15 mL to 50 mg tissue) GTE buffer.

**Isolation of archival nuclei.** We centrifuged prepared tissue for each specimen (4000×*g*, 5 min, 4 °C) and washed with ice-cold PBS. To improve tissue dissociation, we resuspended tissue in 1 mL of pre-heated Pepsin solution (0.5% Pepsin in 5 mM HCl, pH 1.5) and incubated at 37 °C in an Eppendorf ThermoMixer (90 min, 750 rpm). We performed three PBS washes and resuspended in 1 mL sodium citrate buffer (pH 6) before transferring the suspension to a 2 mL glass Dounce homogeniser. For fine tissue dissociation, we homogenised the tissue on ice (approximately 20–30 strokes). To improve nuclei isolation, we performed initial de-crosslinking by incubating at 80 °C in the sodium citrate buffer for 1 h. We then washed the tissue three times with PBS before resuspending in 1 mL ice-cold Farnham Lysis Buffer (FLB; 5 mM PIPES pH 8.0, 0.1% SDS, 1 mM EDTA) and transferring to a 1 mL Covaris milliTUBE containing an AFA fibre. We isolated nuclei by NEXSON[46] through sonication of the tubes in a Covaris E220 focused-ultrasonicator on settings PIP 160, DF 15%, CBP 200 for 600 s. With the mouse samples, we split the tissue in half to process with MNase and FAIRE. With all tubes, we pelleted the nuclei, removed the supernatant, and froze the pellets at −80 °C.

**Archival MNase treatment.** We based the archival MNase protocol on previously published methods[31] with substantial optimisation for heavy fixed input. We resuspended frozen nuclei in 200 μL MNase DB per 50 mg tissue and added 0.5 U MNase (Worthington Biochemical Corporation) and 200 U Exonuclease III (New England Biolabs) per 50 mg of tissue and incubated at 37 °C with 750 rpm rotation for 15 min. We quenched digestion by adding 50 μL QS per 50 mg of tissue and incubating on ice for 10 min. To enhance the release of the digested chromatin from the nuclear debris, we transferred the suspension to a 1 mL Covaris miliTUBEs containing an AFA fibre and briefly sonicated the samples in a Covaris E220 focused-ultrasonicator with settings PIP 160, DF 15%, CBP 200 for 60 s. We clarified by centrifugation (10 min, 9600×*g*, 4 °C) and transferred the supernatant to a new tube. We added 1 μL RNaseA and incubated for 30 min at room temperature, followed by addition of 2 μL 20 mg/mL proteinase K and incubation at 55 °C for 1 h. We purified DNA fragments by phenol:chloroform:isoamyl alcohol (25:24:1) extraction with back-extraction of the organic phase with addition of 150 μL 10 mM Tris, pH 8.0, followed by an additional extraction. We de-crosslinked overnight with incubation at 65 °C and concentrated the DNA via small fragment-optimised bead purification, resuspending in 20 μL 10 mM Tris, pH 8.0.

**Archival FAIRE treatment.** We based the archival FAIRE protocol on previously published methods[25] with modifications to chromatin shearing and extractions. We resuspended nuclei in 1 mL chromatin shearing buffer (10 mM Tris-HCl pH 8.0, 0.1% SDS, 1 mM EDTA) and transferred to 1 mL Covaris milliTUBEs containing an AFA fibre. We sheared chromatin via sonication in a Covaris E220 focused-ultrasonicator with settings PIP 420, DF 30%, CBP 200 for 10 min. We clarified the lysate by centrifugation (5500×*g*, 10 min, 4 °C) and removed the supernatant to a new tube. We added 1 μL RNaseA and incubated for 30 min at room temperature and then reserved 10% of the sheared chromatin as an input control. With the FAIRE fraction, we depleted protein-bound DNA via phenol:chloroform:isoamyl alcohol extraction with back-extraction followed by an additional extraction. To the reserved input control, we added 2 μL proteinase K solution and incubated at 65 °C for 1 h and de-crosslinked both the FAIRE and input fractions overnight with incubation at 65 °C. We performed two phenol:chloroform:isoamyl alcohol extractions with back-extraction upon the input controls and concentrated FAIRE and input fraction DNA via small fragment-optimised bead purification, resuspending in 20 μL 10 mM Tris, pH 8.0.

## Fresh vertebrate tissue processing

**Fixation of fresh tissue.** On dry ice, we transferred approximately 30 mg of liver tissue per flash-frozen specimen to an extra thick TT05 TissueTube and cryo-pulverised into a rough powder using a cryoPREP (Covaris) automated dry pulveriser (two impacts on intensity setting 6). We resuspended the pulverised tissue in 1.5 mL room temperate PBS, transferred to a 2 mL tube and immediately added 40 μL 37% formaldehyde (1% final concentration), followed by rocking at room temperature for 15 min and then quenching fixation by adding 79 μL 2.5 M glycine to a final concentration of 125 mM glycine. We rocked for 5 min at room temperature and then centrifuged (500×*g*, 5 min, 4 °C).

**Isolation of nuclei from fixed frozen tissue.** We washed the pulverised fixed tissue three times with ice-cold PBS, resuspended in 1 mL FLB and transferred to a 1 mL Covaris milliTUBE containing an AFA fibre. We isolated nuclei by NEXSON in a Covaris E220 focused-ultrasonicator with settings PIP 150, DF 10%, CBP 200 for 300 s. We split the tissue in half to process with MNase and FAIRE, pelleted the nuclei, removed the supernatant, and froze the pellets at −80 °C.

**Fixed frozen tissue MNase treatment.** For MNase treatment of the fixed frozen mouse tissues, we adapted published methods[31] to conform with the equipment used in our modified archival MNase protocol. To pelleted nuclei, we added 200 μL DB and gently resuspended. To each tube, we added 0.5 U MNase (Worthington Biochemical Corporation) and 200 U Exonuclease III (New England Biolabs) and incubated at 37 °C with 750 rpm rotation for 15 min. To quench digestion, we added 50 μL QS, mixed well and incubated on ice for 10 min. For enhanced release of digested chromatin, we transferred the suspension to a 1 mL Covaris miliTUBE containing an AFA fibre and briefly sonicated the samples in a Covaris E220 focused-ultrasonicator with settings PIP 160, DF 15%, CBP 200 for 60 s. We transferred the sonicated digest to a new 2 mL tube and clarified by centrifugation (10 min, 9600×*g*, 4 °C), transferring the supernatant to a new tube. We added 1 μL RNaseA and incubated for 30 min at room temperature, followed by addition of 2 μL 20 mg/mL proteinase K and incubation at 55 °C for 1 h. We purified DNA fragments with a phenol:chloroform:isoamyl alcohol (25:24:1) extraction including back-extraction, followed by an additional extraction. We de-crosslinked overnight with incubation at 65 °C and concentrated the DNA via small fragment-optimised bead purification, resuspending in 20 μL 10 mM Tris, pH 8.0.

**Fixed frozen tissue FAIRE treatment.** Following an established FAIRE protocol[25], we processed fresh tissues with equipment modifications. To the pelleted nuclei, we added 1 mL CSB, resuspended and transferred the suspension to new 1 mL Covaris milliTUBEs containing an AFA fibre. We sheared chromatin via sonication in a Covaris E220 focused-ultrasonicator with settings PIP 420, DF 30%, CBP 200 for 12 min. We clarified the lysate by centrifugation (15 min, 5500×*g*, 4 °C), removing the supernatant to a new tube. We added 1 μL RNaseA and incubated for 30 min at room temperature. We reserved 10% of the sheared chromatin to purify as an input control and further processed the FAIRE and input controls as we did the archival samples.

**RNA extraction from fresh mouse tissues.** We extracted RNA from fresh mouse using an AllPrep DNA/RNA kit (Qiagen). We placed 5–10 mg tissue into 350 μL RLT Plus Buffer in a 2 mL tube with a 5 mm stainless steel bead and homogenised with a TissueLyzer (two 2 min rounds, 30 Hz). We then followed the manufacturer's instructions to isolate RNA, eluting in 30 μL RNase-free water.

**Preparation of water dragon input control.** The five archival specimens lacked sufficient archival tissue for processing of an input control and were not archived with specimen-matched fresh tissue. Thus,

we opted to procure fresh tissue from three separate individuals to serve as a pooled input control. We dissected frozen liver tissue from animals euthanised due to injury in accordance with Queensland Department of Environment and Sciences permit WA0038029 (Australian Ethics Committee number ANA20161, University of Sunshine Coast). Using a TissueLyzer, we pulverised approximately 5 mg tissue per specimen in a 2 mL tube containing a 5 mm stainless steel bead. We then immediately added 350 μL RLT Plus Buffer and followed the manufacturer's instructions for the Qiagen AllPrep kit, eluting in 100 μL Elution Buffer.

## Nucleic acid quantification

We quantified DNA by Qubit (1X dsDNA HS Assay kit) and Tapestation (High Sensitivity D1000), per the manufacturer's instructions. We quantified RNA by NanoDrop and Bioanalyzer (Total RNA Nano), per the manufacturer's instructions. We report DNA yield from the vertebrate tissue samples in Tables 3–5.

## Library preparation and sequencing

The Australian Genome Research Facility (AGRF) performed library preparation and sequencing. AGRF prepared all DNA libraries with the xGen cfDNA & FFPE DNA Library Prep Kit (IDT) and sequenced the yeast DNA libraries on a single 200 cycle (100 bp PE) Illumina NovaSeq S4 lane, all mouse and water dragon DNA libraries across four 300 cycle (150 bp PE) Illumina NovaSeq S4 lanes. AGRF prepared Illumina stranded mRNA libraries from the mouse RNA extracts and sequenced the pool on a 100 cycle (100 bp SE) NovaSeq S4 lane.

## Analyses

**Genome preparation.** We used the S288C R64/sacCer3 RefSeq assembly for *S. cerevisiae*, the GRCm38.p6 RefSeq assembly for *M. musculus*, and the *Pogona vitticeps* pvi1.1 RefSeq assembly for *I. l. lesueurii* due to the lack of a species-specific reference. We performed repeat masking with RepeatMasker v.4.1.0 (http://www.repeatmasker.org) through two rounds of masking, first with default settings and with standard Illumina adaptors using -e rmblast enabled.

**DNA read alignment.** We computed quality control metrics for raw reads using FastQC version 0.11.8[47]. For de-duplication with the IDT library unique molecular identifiers (UMIs), we converted the Fastq files to BAM format using FastqToSam (PICARD[48] v 2.9.2), extracted the UMIs with ExtractUmisFromBam (FGBio[49] v. 1.3.0) and restored the files to FastQ format with SamToFastq (PICARD). We aligned raw reads using kalign (ngskit4b tool suite[50] version 200218) with options -c25 -l25 -d50 -U4. We removed PCR and optical duplicates using MarkDuplicates (PICARD) enabling REMOVE_DUPLICATES = TRUE and utilising UMIs.

**Table 4 | Extraction and sequencing details for the fresh mouse specimens**

| Mouse specimen | DNA yield (ng/mg tissue) | Raw read pairs (M) | Reads mapping (%) | Mean insert size (bp) | Mean genome coverage (X) |
|---|---|---|---|---|---|
| Lab1 | 28 | 282 | 82 | 178 | 29.2 |
| Lab2 | 52 | 293 | 81 | 101 | 16.9 |
| Lab3 | 45 | 249 | 83 | 97 | 14.2 |
| Wild1 | 44 | 306 | 84 | 115 | 21.1 |
| Wild2 | 21 | 262 | 93 | 124 | 21.4 |
| Wild3 | 77 | 256 | 85 | 128 | 19.7 |

Summary of the extraction and sequencing results for processing of 30 mg of fresh liver tissue from six mouse specimens with MNase treatment. We report DNA yield in ng/mg of tissue, the number of raw read pairs (millions), the percentage of raw reads mapping to the reference genome, the mean insert size of mapped reads in base pairs and the mean genome coverage after de-duplication and GC correction.

**Table 5 | Extraction and sequencing details for the archival mouse specimens**

| Mouse specimen | Tissue mass (mg) | DNA yield (ng/mg tissue) | Raw read pairs (M) | Reads mapping (%) | Mean insert size (bp) | Mean genome coverage (X) |
|---|---|---|---|---|---|---|
| Lab1 | 287 | 14.4 | 250 | 67 | 129 | 15.3 |
| Lab2 | 545 | 7.8 | 199 | 68 | 124 | 11.9 |
| Lab3 | 593 | 7.6 | 228 | 65 | 124 | 13.1 |
| Wild1 | 467 | 4.5 | 239 | 67 | 117 | 13.3 |
| Wild2 | 624 | 6.8 | 263 | 67 | 118 | 14.9 |
| Wild3 | 763 | 3.5 | 227 | 68 | 112 | 12.2 |

Summary of the extraction and sequencing results for processing available archival liver tissue from six mouse specimens with MNase treatment. We report tissue mass processed in mg, DNA yield in ng/mg of tissue, the number of raw read pairs (millions), the percentage of raw reads mapping to the reference genome, the mean insert size of mapped reads in base pairs and the mean genome coverage after de-duplication and GC correction.

We computed and corrected for GC-bias with deepTools[51] version 3.5.1 using effective genome sizes of 12,157,105 bp, 2,818,974,548 bp and 1,716,675,060 bp and for the sacCer3, GRCm38.p6 and pvi1.1 genomes, respectively. We calculated mean aligned insert length using CollectInsertSizeMetrics (PICARD) and estimated nuclear genome coverage as the number of unique aligned GC-corrected reads multiplied by the mean insert length divided by unmasked genome size. We report sequencing yield and mapping results of all vertebrate samples in Tables 3–5. We visualised alignments in CLC Genomics Workbench 21 (Qiagen).

**Peak analyses.** We analysed regional occupancy after quantile normalisation of sequencing depth differences between treatment and control to calculate differential occupancy values with the dtriple function in DANPOS3[27]. We analysed all yeast and mouse alignments both as individual samples and as pools of three replicates. For profiling the effect of FAIRE and MNase treatment, we used a corresponding input control. For differential peak analyses, we ran pooled heat-shocked cultures versus pooled optimal growth conditions as input control for yeast and pooled laboratory strain versus pooled wild caught as input for mouse. We analysed all water dragon alignments individually compared to a pool of three input control alignments. With the output of the DANPOS dpeak function, we enforced a significance cut-off of FDR < 0.05 upon sites with local peak gains, local peak losses and log2fold-change in total peak signal. We used the R packages *ChIPseeker*[52,53], *GenomicFeatures* and *GenomicRanges*[54] to annotate sites with significant peak changes and restrict our downstream analyses to the 2 kb region upstream of TSSs.

**RNA-Seq analysis.** We aligned RNA-Seq reads from yeast and mouse to their respective masked genomes with kalign and calculated FPKM using the Tuxedo pipeline[55]. We then z-transformed the FPKM values using the R package *zFPKM*[56] and created lists of genes with low (zFPKM < −2) medium (−2 > zFPKM < 2) and high (zFPKM > 2) expression. To account for regions with poor mapping, we eliminated genes from the low score list with Z-scores less than −20 (indicating reads mapped to the gene but at very low levels). For the differential expression analyses, we used the Trinity edgeR pipeline[57,58] to calculate log2FC between treatments.

**Downstream analyses of peak profiles.** To summarise shared genes with significant peak changes between replicates, we used the R package *ggVennDiagram*. We calculated Gene Ontology (GO) Biological Process enrichment for yeast within lists of genes with significant peak changes as well as a list of differentially expressed genes (as determined by RNA-Seq) using Yeast EnrichR[29]. For

mouse, we calculated tissue-specific enrichment within lists of genes with significant peak changes using TissueEnrich[59]. We generated genome-wide TSS peak enrichment heatmaps with the tagHeatmap function in *ChIPseeker*. To measure the effect of gene expression on upstream peak changes, we calculated the proportion of genes with low, medium and high expression with significant peak changes in the 2 kb upstream region for each replicate and plotted the proportions as violin plots.

**Principal component analyses.** To compare genome-wide MNase profiles between individuals for mouse and water dragon, we converted each individual's DANPOS dpeak wig file from having compared the MNase profile to input control to bigWig format and compiled a summary matrix of Pearson correlation values for each species using Deeptools[51]. We performed principal component analysis upon the resulting summary matrices with the *prcomp* function in base R and plotted principal components one through three using *ggplot2*.

**Reporting summary**
Further information on research design is available in the Nature Portfolio Reporting Summary linked to this article.

## Data availability

The raw sequencing data generated in this study have been deposited in the CSIRO Data Access Portal (https://data.csiro.au/) in collections 54007 (mouse data; https://data.csiro.au/collection/csiro:54007), 51669 (yeast; https://data.csiro.au/collection/csiro:51669) and 59757 (eastern water dragon; https://data.csiro.au/collection/csiro:59757). Both the raw sequencing data and processed data (.wig) files have been deposited in the Gene Expression Omnibus database under accession numbers GSE256156 (yeast FAIRE and MNase; https://www.ncbi.nlm.nih.gov/geo/query/acc.cgi?acc=GSE256156), GSE256160 (yeast RNA-Seq; https://www.ncbi.nlm.nih.gov/geo/query/acc.cgi?acc=GSE256160), GSE261169 (mouse FAIRE and MNase; https://www.ncbi.nlm.nih.gov/geo/query/acc.cgi?acc=GSE261169), GSE256158 (mouse RNA-Seq; https://www.ncbi.nlm.nih.gov/geo/query/acc.cgi?acc=GSE256158), and GSE256157 (water dragon; https://www.ncbi.nlm.nih.gov/geo/query/acc.cgi?acc=GSE256157). Source data are provided with this paper as a Source Data file.

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

## Acknowledgements

We thank Olly Berry and Andrew Young for their leadership within the Environomics Future Science Platform. We thank the director of the Australian National Wildlife Collection, Leo Joseph, and the ANWC staff (specifically, Margaret Cawsey, Alex Drew, Tonya Haff, and Chris Wilson) for their contributions of curatorial expertise, metadata management and sampling assistance. We thank Wendy Ruscoe for acquiring wild Australian mice and Oliver Mead for supplying yeast cultures. We thank Patrick Couper and Andrew Amey of the Queensland Museum for permission and assistance in sampling the eastern water dragons. We thank Ondrej Hlinka and CSIRO IM&T Client Services for their assistance in utilising the CSIRO supercomputing system. We thank the Australian Genome Research Facility for sequencing advice. We thank Don Gardiner, Kerensa McElroy, Yijin Liew, Cheng Soon Ong and the Environomics Epigenetics Discussion Group for comments on the study design and analysis. We would like to acknowledge the contribution of Bioplatforms Australia in the generation of data used in this publication. Bioplatforms Australia is enabled by NCRIS. Funding for this study was provided by the Environomics CSIRO Future Science Platform (grants R-10011 and R-14486) awarded to Clare E. Holleley.

## Author contributions

CEH, JT, and EEH conceived of the research approach and designed the project. CHF, NJ, EEH, and MA collected and preserved specimens. EEH, AG, and MA conducted adaptation of the methods for archival specimens and carried out laboratory experiments. EEH, JS, and JT conducted data analysis and data management. EEH produced the figures. EEH and CEH wrote the first draft of the manuscript. All authors discussed the analyses and reviewed the manuscript.

## Competing interests

The authors declare no competing interests.
