## [Peer Review File · Nature Communications]

Century-old chromatin architecture revealed in formalin-fixed vertebratesReviewers' Comments:

Reviewer #1:

Remarks to the Author:

Summary:

I have reviewed the manuscript by Hahn et al., which illuminates chromatin architecture from archived museum lizard specimens after thorough protocol optimization and data exploration in well-controlled yeast/mouse experimental systems. The authors leveraged the fact that numerous modern genomic approaches and long-term practices in museum science both utilize formaldehyde, leading to the hypothesis that chromatin architecture can be recovered from museum specimens. The authors have convincingly shown this hypothesis to be correct and have amassed other interesting and useful results. Specifically, the authors demonstrate the ability to reproducibly unveil chromatin profiles for century-old museum specimens, detecting tissue-specific, sex-specific, and environment-specific profiles. The authors also demonstrate that over-fixation may enable semi-quantitative estimates of gene expression. Altogether, I found that the analyses conducted by the authors were sound, the manuscript was well written, and the results and their broader ramifications will likely be interesting to researchers in the fields of museum science and genomics. I only outline some relatively minor issues or points of clarification. Once reviewer critiques are addressed, I believe this manuscript would be suitable for publication and would be an impactful contribution to the museum science and genomics literature. Personally, I am excited by the results presented in this manuscript and believe they will catalyze epigenomic investigations using museum specimens, which have been largely non-existent so far. Well done!

Daren Card, PhD

Computational Biologist at the Broad Institute and Dana-Farber Cancer Institute

Comments:

1. The authors note that they created an experimental collection of formalin-preserved inbred C57 Black 6 lab mice and outbred wild-trapped mice with flash-frozen liver tissue. I commend the authors for their foresight of the needs of downstream researchers and for setting an example of the types of "museum genomics" collections that natural history museums could be generating beyond traditional museum collecting approaches. By complementing existing taxonomically or geographically focused collections with methodologically focused collections targeted towards emerging experimental areas like genomics, I believe natural history museums can broaden their impact on modern science. However, I see no mention of the specific museum accessions linked to these voucher/tissue samples, which may impact the ability of future researcher to access them, so I encourage the authors to provide this information in a supplementary table.
2. Line 211: Change "cover" to "coverage".
3. Line 231: The authors correctly note that the interval between death and formalin fixation is rarely recorded. However, they missed the opportunity to also point out that the fixation time (better would be fixation time per gram or some other standardized measure) is also rarely, if ever, recorded. Both of these pieces of information, and potentially others, will be invaluable for applying approaches like that laid out in this manuscript to museum specimens. I think the authors could have more forcefully pointed these considerations out and encouraged museum practitioners to adopt more robust experimental data reporting during specimen/tissue collection, at least for future collection.
4. Line 521: Change "dontaed" to "donated".
5. Figure 1E: The authors compared GO terms from up-regulated RNA-seq genes and genes with nearby MNase signal gain, finding some overlap. However, when GO terms have enough corresponding genes, it is possible that similar enrichment could be observed from largely non-overlapping gene sets. Is the overlap of the genes that were used for these GO enrichment analyses correspondingly high? A simple Venn diagram would summarize this. The authors could also consider a random permutation approach that controls for their experiment to determine whether this overlap is greater than what would be expected by chance, which would bolster this conclusion quantitatively.
6. Figure 4 Caption: In the figure, you use the term "accessible" but in the caption, I instead see

"assessable" on line 422. I'm assuming this is a typo and wanted to highlight it.

7. Data Availability: I commend the authors on archiving their raw sequencing data prior to manuscript submission! However, I have some additional suggestions for archiving data that I believe will broaden the usefulness of this paper and its data for other investigators. First, I recommend also archiving raw read data in a governmental genetics/genomics repository such as NCBI where it can be queried by others, as this repository, while useful, is not as widely known/searched as existing repositories. The NCBI SRA, for example, also tracks important metadata about the samples/sequencing runs better. The CSIRO repository is useful as a backup and I also encourage the authors to consider archiving their downstream mapping BAM files and/or peak files, which other investigators may find useful. Finally, in the case of the yeast repository on NCBI, there are no nested directories or README files that allow others to understand which file corresponds with which experimental condition, so these details should be added so that these data can be used by others (the mouse/lizard data do not have this problem).

8. Extended Data Figure 3: Unfortunately, the resolution of these figures is low and some of them cannot be properly interpreted (especially the Venn diagrams in B). Fortunately, the colors allowed me to crudely interpret these figures, but I encourage the authors to use higher resolution figures to avoid this issue.

Reviewer #2:

Remarks to the Author:

Linking the fields of chromatin biology with museomics is a novel approach and one that can be useful for future applications on both sides of the topic. This was an exciting paper to read. To my knowledge, such a study has never been previously attempted and as the usage of museum collections becomes more sophisticated in what can be learned, studies such as this accelerate that usage. As someone who works with formalin-fixed specimens for molecular applications, I find this study very noteworthy.

Given that the time-series with the mice is pretty short term (which there is no way to avoid in this type of stuff), are there any concerns about whether the results are applicable/expected to be interpreted the same way for 100yo lizard specimens given there is no control/comparison of fresh tissue for it, as with the mice? This should be mentioned as to why/why not this matters (although Fig 5 makes it look like you expect and get the same types of info and results, which is great!).

A small note: in reviewing Ruane and Austin 2017, one of the snakes sequenced was collected between 1878 and 1911, so I would hedge the statement made in line 209-210 as it may not be wholly accurate as written/interpreted by the reader, revise slightly.

In introducing the topic, the authors do an excellent job placing the usage of formalin in context with museum collections and providing relevant background and citations. With this in mind, I would suggest in the discussion slightly expanding on how these novel methods can be used across multiple components of biology related to museum collections science---e.g., systematics, evolutionary history, disease detection, etc.

Many people who are using formalin-fixed specimens for molecular applications will be very interested in this paper, but many of those working with museum collections do not ever work with chromatin specifically. Adding a little bit more of an extension to the uses highlighted and potentially resulting from this work to the museum-user community and/or how this can be applied in multiple ways/uses would be excellent and make the paper even more appealing. There is a bit of this in the discussion but I think it could be lengthier as I think it is the most interesting for readers thinking about their own work.

The figures look excellent and are easy to follow.

I look forward to seeing this in print.

RESPONSE TO REVIEWER COMMENTS

Reviewer #1 (Remarks to the Author):

Summary:

I have reviewed the manuscript by Hahn et al., which illuminates chromatin architecture from archived museum lizard specimens after thorough protocol optimization and data exploration in well-controlled yeast/mouse experimental systems. The authors leveraged the fact that numerous modern genomic approaches and long-term practices in museum science both utilize formaldehyde, leading to the hypothesis that chromatin architecture can be recovered from museum specimens. The authors have convincingly shown this hypothesis to be correct and have amassed other interesting and useful results. Specifically, the authors demonstrate the ability to reproducibly unveil chromatin profiles for century-old museum specimens, detecting tissue-specific, sex-specific, and environment-specific profiles. The authors also demonstrate that over-fixation may enable semi-quantitative estimates of gene expression. Altogether, I found that the analyses conducted by the authors were sound, the manuscript was well written, and the results and their broader ramifications will likely be interesting to researchers in the fields of museum science and genomics. I only outline some relatively minor issues or points of clarification. Once reviewer critiques are addressed, I believe this manuscript would be suitable for publication and would be an impactful contribution to the museum science and genomics literature. Personally, I am excited by the results presented in this manuscript and believe they will catalyze epigenomic investigations using museum specimens, which have been largely non-existent so far. Well done!

Daren Card, PhD

Computational Biologist at the Broad Institute and Dana-Farber Cancer Institute

Response: We thank Reviewer 1 for their overwhelmingly positive review and agree that this technique has the potential to spawn a whole new field of research focussing on the temporal and evolutionary dynamics of gene regulation. There is an urgent need in the scientific community to understand base-line historical responses to the environment that occurred prior to contemporary climate change. We sincerely hope to inspire a range of techniques that enable investigations to establish the trajectory of gene regulation change.

Action: None required

Comments:

1. The authors note that they created an experimental collection of formalin-preserved inbred C57 Black 6 lab mice and outbred wild-trapped mice with flash-frozen liver tissue. I commend the authors for their foresight of the needs of downstream researchers and for setting an example of the types of “museum genomics” collections that natural history museums could be generating beyond traditional museum collecting approaches. By complementing existing taxonomically or geographically focused collections with methodologically focused collections targeted towards emerging experimental areas like genomics, I believe natural history museums can broaden their impact on modern science. However, I see no mention of the specific museum accessions linked to these voucher/tissue samples, which may impact the ability of future researcher to access them, so I encourage the authors to provide this information in a supplementary table.

Response: We agree with Review 1's assessment regarding the importance of specifically designed and accessioned collection material that is destined for future methodological focused research. This is indeed a new type of collection object with a different set of goals and applications. We have amended the manuscript to highlight the novelty and importance of this approach future with emerging technologies.

Actions:

Lines 363 – 374: Added text to the discussion highlighting the purpose of creating a methodologically-focused collection.

“To develop methods tailored to the challenges posed by archival specimens, we created a bank of mock-preserved specimens with matched frozen tissue and extensive preservation metadata. We encourage the world's museum collections to follow suit and prepare methodologically-focused collections for the purpose of enabling future research. Further to that, we recommend inclusion of preservation history details among specimen metadata to enable accurate specimen suitability screening. Doing so will ensure that the methods described here become part of a suite of approaches to characterise historical genomes and gene regulation.”

Lines 472 – 484: Added text to the methods to elaborate on the preparation of the experimental collection of mice.

“To enable comprehensive methods development for this and future studies, we created a bank of experimental specimens representative of the wide variety of liquid preservation practices used in collections over the last century. We selected a species with a well-annotated genome and tractable tissue volume for processing and storage, *Mus musculus* (strain C57BL/6). We acquired 151 male mice aged 17-18 weeks from Australian BioResources and sacrificed them upon arrival by cervical dislocation (Australian Ethics Committee number 2017-34). In accordance with modern archival procedure, we sampled liver tissue from specimens for storage at -80°C as specimen-matched fresh tissue. We then processed the intact carcasses using several preservatives (unbuffered formalin, neutral buffered formalin, and formalin acetic acid) with fixation times ranging from 15 minutes to 3 days with or without washing with water and stepping into ethanol for long-term storage. For all mice used in this study, we immersed each carcass in 10% neutral buffered formalin (3.7% formaldehyde) for three days followed by soaking in water for one day before transfer to 70% ethanol.”

Lines 493 – 495: Added text to directly point to where the accession numbers can be found in Table 1. We have also moved the tables from the Supplement to the main article for easier access.

“We archived all mouse specimens in the Australian National Wildlife Collection (ANWC; Crace, ACT, Australia) spirit vault in glass specimen jars (see Table 1 for registration details of specimens used in this study).”

2. Line 211: Change “cover” to “coverage”.

Action: amended, now line 320.

3. Line 231: The authors correctly note that the interval between death and formalin fixation is rarely recorded. However, they missed the opportunity to also point out that the fixation time (better would be fixation time per gram or some other standardized measure) is also rarely, if ever,

recorded. Both of these pieces of information, and potentially others, will be invaluable for applying approaches like that laid out in this manuscript to museum specimens. I think the authors could have more forcefully pointed these considerations out and encouraged museum practitioners to adopt more robust experimental data reporting during specimen/tissue collection, at least for future collection.

Response: We completely agree with reviewer one and we have added information about the additional challenges faced by unknown fixation conditions to the introduction. We also included new recommendations in the discussion of useful metadata fields relevant to this technique, going forward. By including best practise record keeping for this specimen characteristic/s, we hope to influence and improve the utility of contemporary collections for future epigenomic applications.

Actions:

Lines 107 –109: Added text to point to additional challenges.

“Additional challenges exist because fixation conditions for museum specimens are rarely (if ever) recorded in specimen metadata (e.g., fixation time/volume per gram), resulting in specimens having unknown preservation states.”

Lines 366-368: Added text recommending inclusion of fixation procedures in specimen metadata.

“Further to that, we recommend inclusion of preservation history details among specimen metadata to enable accurate specimen suitability screening.”

4. Line 521: Change “dontaed” to “donated”.

Action: amended, now line 485.

5. Figure 1E: The authors compared GO terms from up-regulated RNA-seq genes and genes with nearby MNase signal gain, finding some overlap. However, when GO terms have enough corresponding genes, it is possible that similar enrichment could be observed from largely non-overlapping gene sets. Is the overlap of the genes that were used for these GO enrichment analyses correspondingly high? A simple Venn diagram would summarize this. The authors could also consider a random permutation approach that controls for their experiment to determine whether this overlap is greater than what would be expected by chance, which would bolster this conclusion quantitatively.

Response: This is another excellent suggestion that we have used to strengthen the revised version of our manuscript. We are grateful to Reviewer 1 for suggesting this analysis. As requested, we have added a Venn diagram to Figure 1 to illustrate the overlap in all genes used for the GO term analysis. We also conducted the suggested permutation test and can report that our shared number of genes (108) is significantly higher (adjusted $p = 6.061912e-87$) than the mean number of shared genes between two randomised lists of the same size as our gene lists (mean 19.989, std deviation 4.37) permuted 1000 times. Thus, we can confidently conclude MNase chromatin profiles and RNA-seq data sets are both reporting heat shock response signals of gene expression.

Actions:

Figure 1: Venn diagram added as panel E

Lines 172 – 175: Text added to report the results of the permutations test as follows:

“Of the 383 genes with MNase signal gain after 24-hour fixation, 108 were also identified as upregulated by RNA-Seq, which is significantly higher than the mean number of genes shared between two random gene sets across 1000 permutations (Figure 1E; mean = 19.989, standard deviation = 4.37, adjusted p-value = 6.061912^{-87}).”

6. Figure 4 Caption: In the figure, you use the term “accessible” but in the caption, I instead see “assessable” on line 422. I’m assuming this is a typo and wanted to highlight it.

Action: amended, now line 1030.

7. Data Availability: I commend the authors on archiving their raw sequencing data prior to manuscript submission! However, I have some additional suggestions for archiving data that I believe will broaden the usefulness of this paper and its data for other investigators. First, I recommend also archiving raw read data in a governmental genetics/genomics repository such as NCBI where it can be queried by others, as this repository, while useful, is not as widely known/searched as existing repositories. The NCBI SRA, for example, also tracks important metadata about the samples/sequencing runs better. The CSIRO repository is useful as a backup and I also encourage the authors to consider archiving their downstream mapping BAM files and/or peak files, which other investigators may find useful. Finally, in the case of the yeast repository on NCBI, there are no nested directories or README files that allow others to understand which file corresponds with which experimental condition, so these details should be added so that these data can be used by others (the mouse/lizard data do not have this problem).

Response: We agree with the reviewer on the benefit of archiving the raw data and processed files through NCBI to better align with FAIR data standards. We have since archived our raw sequencing data as well as the processed WIG files for each of the datasets through the NCBI GEO. These files are now publicly available, and accessions are provided in the manuscript. We also thank the reviewer for catching the issue with the yeast data on the DAP, allowing us to rectify this oversight.

Actions:

Lines 726 – 729: Added accession numbers for the datasets now available through the NCBI GEO.

We have also restructured the yeast dataset on the DAP to include nested directories and have added a sample key to the collection.

8. Extended Data Figure 3: Unfortunately, the resolution of these figures is low and some of them cannot be properly interpreted (especially the Venn diagrams in B). Fortunately, the colors allowed me to crudely interpret these figures, but I encourage the authors to use higher resolution figures to

avoid this issue.

Action: For the Extended Data Figures, now in the Supplement, we have increased their resolution for submission in a PDF. The main article figures are also provided in high resolution .AI format.

Reviewer #2 (Remarks to the Author):

Linking the fields of chromatin biology with museomics is a novel approach and one that can be useful for future applications on both sides of the topic. This was an exciting paper to read. To my knowledge, such a study has never been previously attempted and as the usage of museum collections becomes more sophisticated in what can be learned, studies such as this accelerate that usage. As someone who works with formalin-fixed specimens for molecular applications, I find this study very noteworthy.

Response: We thank reviewer 2 for their enthusiasm, and for highlighting the novelty of our work and how we are pushing the boundaries of molecular technologies tailored for the unique properties of museum specimens.

Actions: none required.

#1 Given that the time-series with the mice is pretty short term (which there is no way to avoid in this type of stuff), are there any concerns about whether the results are applicable/expected to be interpreted the same way for 100yo lizard specimens given there is no control/comparison of fresh tissue for it, as with the mice? This should be mentioned as to why/why not this matters (although Fig 5 makes it look like you expect and get the same types of info and results, which is great!).

Response: As reviewer 2 points out, we expect to observe the same signal of peak enrichment around TSS (chromatin accessibility) regardless of specimen age. This is demonstrated in several figures within the paper where a “funnel” of enrichment centred around the TSS, can be observed in all vertebrate specimens that we have analysed and visualised with the DANPOS Heatmaps – see what is now Supplementary Figure 8 (all individual water dragon heatmaps) and newly added Supplementary Figures 4 & 5 (all individual mouse heatmaps) and the right hand column of figure 4 “archival” fixation (combined dataset heatmaps for each species). The robust preservation of chromatin accessibility signatures during extended preservation and storage times is indeed one of the strengths of our approach. However, specimen quality, which can be a function of age, does affect other data characteristics, such as fragment length distribution, proportion of reads mapped, and library complexity assessed as the % of duplicate reads. These characteristics are consistent with observations across museum genomics and ancient DNA research and do not affect the power of our analysis if compensated by increased sequencing effort, assuming the sample is of sufficient quality for sequencing.

Actions:

Lines 305 - 308: Added text suggesting future studies target somewhat older specimens with matched frozen tissue to confirm our results. These lines now read:

“Future studies may also consider strategic selection of somewhat older formalin-preserved specimens having corresponding frozen tissue to confirm the correspondence of the archival chromatin signal with that from fresh tissue in specimens older than our relatively recently preserved mice.”

We added two new supplementary figures (now Supplementary Figures 4 & 5) to display the individual mouse heatmaps and underscore the assay’s repeatability and signal similarity between these more recently preserved mice and the older water dragon samples.

We edited the extended data figure containing the individual water dragon heatmaps (now Supplementary Figure 8) and its legend to include additional information about the sequencing effort applied to each sample (raw reads, mapping rate, and mean genome cover).

#2 A small note: in reviewing Ruane and Austin 2017, one of the snakes sequenced was collected between 1878 and 1911, so I would hedge the statement made in line 209-210 as it may not be wholly accurate as written/interpreted by the reader, revise slightly.

Response: Thank you very much pointing this out, it gives us an opportunity to explain and clarify our novelty. The oldest specimen within Ruane and Austin (2017) (1878-1911) is potentially older than our oldest specimen (1905) but this cannot be determined due to uncertainty in the Ruane and Austin (2017) collection date. However, we actually intended to emphasise that we are the only study to date that has empirically confirmed that the museum specimens have actually been preserved in formalin prior to sequencing them, using formalin quantification test strips. All previous studies have either relied on metadata or simply assumed that wet collection items of a certain age used formalin. Additionally, the Ruane and Austin (2017) work was not whole genome sequencing data, they used a reduced representation sequencing approach (Ultra Conserved Elements) to obtain orthologous gene sets for phylogenetic analysis. We acknowledge that the way this statement was formerly worded was ambiguous and have re-phrased for accuracy.

Action:

Lines 109 – 115: We have added a statement to point out that previous studies have not empirically confirmed formalin fixation of the specimens:

“For this reason, previous studies reporting successful DNA extraction from formalin-preserved museum specimens have either relied on incomplete metadata or simply assumed that wet collection items of a certain age used formalin. By assessing specimen quality directly through visual inspection of the tissue and measurement of media pH and formaldehyde concentration, we have previously shown that formalin-preserved specimens can be identified and vetted for sequencing suitability.”

Line 310 – 312: Now reads:

“Here, the 1905 specimen represents the oldest confirmed formalin-preserved museum specimen to successfully yield genome-wide sequencing data to date ...”

#3 In introducing the topic, the authors do an excellent job placing the usage of formalin in context with museum collections and providing relevant background and citations. With this in mind, I would suggest in the discussion slightly expanding on how these novel methods can be used across multiple components of biology related to museum collections science---e.g., systematics , evolutionary history, disease detection, etc.

Response: See response to Reviewer 2, comment 4 (below) for a combined response to this and the following comment.

#4 Many people who are using formalin-fixed specimens for molecular applications will be very interested in this paper, but many of those working with museum collections do not ever work with chromatin specifically. Adding a little bit more of an extension to the uses highlighted and potentially resulting from this work to the museum-user community and/or how this can be applied in multiple ways/uses would be excellent and make the paper even more appealing. There is a bit of this in the discussion but I think it could be lengthier as I think it is the most interesting for readers thinking about their own work.

Response: We have taken these excellent suggestions from Reviewer 2 to expand in the Discussion on the potential applications of our method within and beyond the museum community.

Lines 378 – 401: Now read:

“Our method is likely to be appropriate for use across vertebrates, with the only requirement being the availability of a species-specific or closely related reference genome. Museum scientists and wildlife researchers may not yet routinely use chromatin accessibility data but should be aware of the new capability this approach adds. For example, systematists can gain both the standard genomic data required for phylogenetic and/or population genomic analysis (e.g. whole mitochondrial genomes, orthologous gene sets, single nucleotide polymorphisms), and simultaneously characterise adaptive responses to the environment. This extra layer of information could help detect species provenance, identify structure in widely distributed vagile populations (e.g. fish stocks) or disambiguate rapid or recently diverged lineages (e.g., introduced and invasive species). Conservation biologists and wildlife managers may use the method to establish historical baselines against which to track rapid adaptation to captivity during ex-situ conservation efforts. Having a quantitative and reproducible molecular approach to characterise phenotypic plasticity allows managers to establish clear decision-making thresholds for species release and red-flag triggers during ex-situ monitoring. There is also the potential to contribute to biosecurity monitoring either by the direct detection of pathogens and/or host responses to infection.

Most importantly, our method provides the unique opportunity to survey gene expression in both modern and historical specimens and reveal the mode, tempo and directionality of gene expression change over the last century. Generation of historical chromatin profiles at scale may enable statistical modelling approaches that leverage past and present data, to predict future state. This provides essential context for estimating the resilience of contemporary populations to future environmental threats and may reveal the full repertoire of rapid evolutionary responses to environmental challenges such as climate change and introduced pests and pathogens.”

Reviewers' Comments:

Reviewer #1:

Remarks to the Author:

I have rereviewed the manuscript by Hahn et al., which illuminates chromatin architecture from archived museum lizard specimens after thorough protocol optimization and data exploration in well-controlled yeast/mouse experimental systems. My prior review of this manuscript was generally positive and I only highlighted some minor points that needed clarification or further work. I also reviewed the author responses to my previous critiques, which were thoughtful and addressed my feedback in full. I therefore affirm that this manuscript is sufficient quality for publication, having improved during the review process, and I believe it would be a great contribution to the literature in the fields of ecology and evolutionary biology, genomics, museum science, and more.

Daren Card, PhD

Computational Biologist at the Broad Institute and Dana-Farber Cancer Institute

Reviewer #2:

Remarks to the Author:

The authors have done an excellent job addressing all of my concerns and suggestions and I have nothing further. I look forward to seeing this paper in print.